# Energy Hogs and Misers: Magnitude and Variability of Individuals' Household Electricity Consumption

Claudia Bustamante [1], Stephen Bird [2], Lisa Legault [3] and Susan E. Powers [1,*]

1 Institute for a Sustainable Environment, Clarkson University, Potsdam, NY 13699, USA
2 Institute for a Sustainable Environment and Political Science, Clarkson University, Potsdam, NY 13699, USA
3 Department of Psychology, Clarkson University, Potsdam, NY 13699, USA
* Correspondence: spowers@clarkson.edu; Tel.: +1-315-268-6542

**Abstract:** We use circuit-level granular electricity measurements from student housing and statistical analysis to better understand individuals' electricity consumption. Two key patterns emerged—individuals varied systematically in their magnitude of electricity use as well as in their variability of usage at the hourly and daily level. A cluster analysis of electricity consumption in individual bedrooms shows that 18% of students consume 48% of total electricity use at a median of 2.17 kWh/d/person. These few energy hogs have a disproportionate impact on electricity consumption. In contrast, the misers (22% of students) consume only 4% of the electricity (0.18 kWh/d/person). Mini-refrigerators in bedrooms contributed substantially to the total electricity use of the moderate users. In contrast, mini-refrigerators were less influential for energy hogs, suggesting that these residents may draw power in others ways, such as by using powerful computing or gaming systems for hours each day. A sub-cluster analysis revealed substantial individual variability in hourly usage profiles. Some energy hogs use electricity consistently throughout the day, while others have specific periods of high consumption. We demonstrate how our analysis is generalizable to other situations where the resident does not directly pay their utility bills and thus has limited financial incentive to conserve, and how it contributes to a deeper understanding of the different ways in which individuals use energy. This allows for targeting interventions to groups with similar patterns of consumption. For example, policies such as fines or fees that might reduce the excessive electricity use for short times or for individual hogs could result in potential savings ranging from 16–33% of bedroom electricity.

**Keywords:** energy conservation; occupant behavior; end-use; residential buildings; university; individual electricity use





## 1. Introduction

In the U.S., the residential sector accounts for 22% of total energy consumption and about 19% of carbon dioxide emissions [1,2]. Federal and state policies have focused on reducing this energy use to decrease fossil fuel consumption, reduce greenhouse gas emissions, and save money for homeowners.

Approaches to tackle this problem involve understanding interconnected technical, physical, and human factors. Technical and physical factors include climate, building envelope, and building infrastructure [3]. Human factors, such as operation and maintenance, indoor environmental conditions, appliance use and occupant activities and behavior, can have a greater impact in reducing energy consumption than technical and physical factors [3]. Similarly, Zhao et al. found that advanced technological building systems contributed to 42% of the potential for energy efficiency, suggesting that 58% of the potential for energy efficiency lies in human-influenced factors [4].

An extensive amount of literature has emerged to understand and respond to human factors in the residential sector, particularly in terms of occupant activities and behavior towards energy conservation. Residents need better information about their utility use [5–9].

This includes information frequency and detail [5,8], associated costs [7], and appliance efficiency [9]. For example, the meta-analysis conducted by [6] found that strategies providing individualized energy audits and consulting are more effective for conservation behavior than strategies that provide historical, peer comparison energy feedback.

Residents' lack of information about their electricity use creates a gap between the degree to which they acknowledge the need for resource efficiency practices and their actual conservation actions [10,11]. This information deficit is compounded for residents who do not see or pay their own utility bills because they are included in a rent payment, or they receive housing at little or no cost. For example, those in residential housing such as university residences, rental units, low-income housing, or army barracks. Consequently, these residents lack both this financial motivation to conserve and the information from monthly bills that can foster awareness of the quantity and consequences of energy consumption habits and potential for conservation [11].

A personalized experience can help users better understand and take action towards energy conservation [12–14]. Therefore, modification of resident behaviors requires that they have tailored information that can help them understand their personal motivation to change a specific behavior [15–17]. We suggest that detailed granular electricity data, which are captured over short time intervals and at the circuit level, can provide a better understanding of different types of users, the magnitude and variability of their patterns of electricity consumption, and the impact of appliances on individual electricity use.

Many previous studies have used electricity consumption data at the household level to assess the type of household and their contribution to overall electricity consumption. For example, Refs. [17,18] used half-hour interval electricity data to understand the electricity use of household customers in subpopulations. Similarly, Satre-Meloy et al. [19] used household electricity data to identify hourly load profiles as a function of the time of day and day of the week. McLoughlin et al. [18] classify households based on diurnal, intra daily, and seasonal variations. Alternately, Ushakova and Mikhaylov [20] classify households based on their daily electricity use. These studies illustrate that heterogeneous households can be characterized based on the magnitude of their electricity use. They also emphasize the importance of disaggregating data to develop detailed electricity use patterns that aggregated data could not identify.

Granular data can quantify consumption behavior and appliance use at the household level. Albert and Rajagopal [21] used smart meters to assess household electricity consumption with an emphasis on determining the magnitude, duration, and variability in use to characterize the household's behavior and identify occupancy of the residence. Smart meters used by Azaza and Fredrik [22] characterized households' responsibility toward and variance in use that contributes to peak loads. Diawuo et al. [23] found that 93% of residential electricity use is associated with seven energy intensive appliances: refrigerator, air conditioner, television, freezer, fan, electric iron, washing machine and compact fluorescent lights. Asensio and Delmas [24] found that the major contributions to electricity consumption in the house are plug load equipment at 36% followed by refrigerators at 19%. Typically, these sorts of household appliances are used by all members of the household. Collectively, these studies identify appliance type and user behavior as a source of variability in the magnitude and hourly load profiles of household electricity use. The papers discussed here use mathematical algorithms to identify particular energy use activities, hourly load profiles, and individual behavior. They generally do not have substantial databases of measured electricity use data to verify their approaches (e.g., [19]). They identify though the need to understand energy use at this level of detail in order to better design intervention strategies, to motivate behavior change or incentivize demand response approaches to reducing or shifting peak loads.

In contrast to studies of household-level electricity use, the measurement of individual level electricity use is rare. Past research has not typically incorporated analysis at granular levels of time (e.g., minute-, hour-, and daily-level data instead of weekly or monthly data) or has not analyzed usage patterns at the level of individual circuits or outlets. The

exception is the measurement in office settings. Murtagh et al. [25] were the first to publish actual individual consumption in a group setting in an office. Using circuit-level outlet data for daily and weekly energy use, these researchers identified the individual usage of 83 participants. They found three types of energy users: typical, always active, and maximal, suggesting a wide range in the magnitude of electricity use among individuals. Similarly, Rafsanjani and Ghahramani [26] make use of electricity meters and sensors to identify individual energy users and their use of shared appliances in an office setting. Ten users were analyzed over a 6-week period and categorized based on their energy use intensity and energy behaviors. They also found a wide range of the magnitude of individual electricity use. In a follow up study [27], sophisticated disaggregation algorithms used to isolate individual electricity use behaviors was touted for its potential cost-effective opportunities for triggering behavior modifications in an office setting.

Other researchers have also studied office settings; Kamilaris et al. [28] use outlet data to identify desktop electricity use for 18 participants. Their study identified potential energy savings linked with usage patterns such as setting sleeping mode at lunchtime and turning off the computer over the weekend. Annaqeeb et al. [29] recorded outlet data at 15 min intervals to analyze the contribution of 8 individual users in an office setting. Their study showed that personal computers, monitors, and lamps contribute to electricity consumption even when individual users are not in the office. They also found differences in usage due to individual work schedules. Understanding individual electricity use at a granular level among members in a residential context has not been studied to the same degree as in office or work contexts. The work presented here is some of the first to do so.

To understand and assess individual residential electricity consumption, the Smart Housing testbed at Clarkson University (Potsdam, NY, USA) was developed. In 2013, the University integrated utility monitoring equipment in four renovated buildings, including circuit-level electricity meters for two of the buildings [30]. The testbed allows highly granular measurements of student utility use in campus apartment housing. Previous results show that group-level education and feedback interventions at the apartment level promote water and electricity conservation behaviors [30]. The results presented here extend that work to focus on the consumption behaviors of individuals and a forthcoming paper correlates these consumption behaviors to their motivational and ideological characteristics.

## 2. Research Scope

### 2.1. Objectives

This paper uses individual residential electricity consumption at a granular level (e.g., hourly and daily use data at the circuit level) to identify individuals who use the most energy, and to identify attributes that characterize different types of users. The research results of this study could improve interventions and policies designed to reduce use, influence energy use behavior, and help residents understand their energy use patterns.

Three primary objectives characterize this analysis:

1. Understand and quantify individual electricity consumption at a granular level in residential settings;
2. Identify and improve our understanding of variables linked with individual-level energy consumption in residential settings;
3. Outline the benefits of these findings to inform energy conservation interventions and policies.

The research presented here focuses on electricity use by individuals rather than in group areas. Having detailed data specifically for bedrooms provides the best available proxy for individual use within multi-resident apartments. The discussion section explores the quality of this assumption. Data from appliances, electronics, and lights plugged into outlets are included in this analysis. Past research suggests that plug loads account for a significant fraction of residential electricity consumption. Moreover, as demonstrated in

Section 4.1, bedrooms comprise the greatest singular use of electricity in the apartments compared to common areas.

This manuscript includes four additional sections that identify the methods (Section 3), results describing electricity use and characterization at the individual level (Section 4), discussion of the implications and use of these findings (Section 5) and conclusions (Section 6).

*2.2. Characteristics of the Smart Housing Testbed Used for This Field Study*

Clarkson's Smart Housing testbed serves as the location for the data and activities that inform our study. We use data from fall semesters over seven years (2013 to 2019; ~80 days per semester). These periods were unaffected by related studies that evaluated interventions or treatments designed to curb individual electricity use [30]. The testbed comprises two buildings with six apartments and a capacity of 28 students in each. All apartments contain single-occupant bedrooms. This results in 84 apartments and a potential for 392 individual bedroom occupants for analysis over the seven-year period.

Data quantifying electricity use (watt-hours per minute) was collected with Triacta PowerHawk 4124 m [31]. These revenue-grade meters are capable of monitoring 24 circuits of single-phase service, which allows one meter to measure the electricity consumed at each circuit of each apartment. The meters are hardwired to electrical lines mounted inside the wall with current transformers (CTs) hardwired to each circuit breaker. Data used in this analysis includes everything plugged into an outlet and overhead lighting in the apartments. No heating, ventilation or air conditioning equipment electricity use was considered.

The database server pulled a reading from each meter at 60-s intervals for storage in a virtual network of servers provided by IBM. The data set was available to the research team via the project's built-in data export tool for analysis over various time scales. All the original, non-manipulated data readings were automatically backed up on separate servers.

Demographic characteristics in our data set involved undergraduate students living on campus with a mean age of $20.5 \pm 1.0$. They were dominated by white (80%), male (53%), and engineering majors (66%). The median reported parental income is less than USD 75,000. These students were randomly chosen by lottery to live in this housing testbed due to the desirability of apartment housing. They pay a flat housing fee each semester that is independent of their personal utility use. The supplemental materials (SM-6) include additional student information.

## 3. Research Methods

Three phases of this work align with the objectives. Details on the data cleaning and statistical analysis are in Sections 3.1 and 3.2.

**Phase 1. Preliminary analysis to identify trends, potential variables, and limitations in the dataset:** We start with an exploratory data analysis using time series, and assessing data distributions for hourly and daily electricity use for all users. We then evaluate and interpret missing data for bedroom use. Data cleaning protocols focus on users who actively use their bedrooms.

**Phase 2. Statistical analysis to identify key variables to characterize electricity use behavior among individuals in their bedrooms**: Descriptive and comparative statistics (Wilcoxon test) were used to explore and identify variables that potentially affect bedroom electricity use patterns. We use a Principal Component Analysis (PCA) to identify the key variables to characterize bedroom electricity use and use cluster analysis to combine users by the main key variable identified by the PCA and to identify a second key variable to further characterize users within each primary cluster. Ultimately, we assess minirefrigerator use as an exploratory analysis of individuals' appliance use and consumption.

**Phase 3. Value of intervention or policy initiatives to reduce electricity consumption:** data collection and sampling are used to analyze potential interventions to reduce electricity use.

### 3.1. Details—Data Cleaning Protocols

Preparing the raw meter data for analysis involved downloading it from the archive server into SPSS (statistical software, IBM, Armonk, NY, USA), applying a series of corrective scripts, determining missing data, and aggregating the data from minute to hour and daily totals. Due to random missing data, aggregated hourly and daily electric energy totals (EE) could not simply be determined as a sum of the available data. An average of the data that was available was assumed to reflect any missing data (Equation (1)). However, if more than 10% of the 60 min in an hour or of 1440 min in a day were missing, the entire hour or day was identified as a missing value and was not included in any of the statistical analyses.

$$EE_{Daily\ Total} \left[ \frac{Wh}{d} \right] = \frac{\sum_{t=1}^{1440} EE_{min} \left[ \frac{Wh}{min} \right]}{Count(t = 1 - 1440)} \times 1440 \left[ \frac{min}{day} \right] \qquad (1)$$

The compiled hourly and daily electric energy dataset was reduced to include days of the fall semester when students were living in their residences and showing active electricity use. Break periods were excluded starting on the Friday before the break and resuming at the end of the official break period. We also deleted other periods when it was apparent that there was no or little activity in the bedroom. This ensured our focus on identifying and characterizing actual individual electricity consumption and behaviors among users.

Determining active occupancy required criteria to evaluate when students were not using their rooms (e.g., absences from campus or staying regularly in another room for a variety of reasons). Of the possible 392 possible bedrooms, 44 were vacant for the semester and removed from the dataset. In addition, bedrooms that appeared vacant were also disqualified based on the number of days a room had zero electricity consumption. Daily electricity use of 0 kWh for more than three days was recorded as blank values. The 10th percentile of the daily electricity consumption among all users was then used to identify and exclude additional days for some residents with low electricity use in the bedroom that suggested inactivity. The data cleaning protocol reduced the population from the 348 bedrooms with assigned residents to 312 inhabited bedrooms with electricity use activity.

### 3.2. Details—Statistical Analyses

Due to the non-normal distribution of electricity use values, we used the non-parametric Wilcoxon signed rank test [32] to determine if the means of two data sets are reliably different. Based on variables identified in the literature, we explored comparisons among years, apartment size (3-, 4-, 6-member apartments), time considerations such as between months, classroom scheduling (Tu-Th versus M-W-F class schedules), and potential differences between weekends and weekdays.

A preliminary exploratory principal component analysis (PCA) conducted with SPSS identified potential variables to characterize individual electricity use. This technique replaces the large set of measured electricity use variables by a smaller number of derived variables, the principle components, while minimizing the loss of information [33]. The derived variables are linear functions of the original variables, with the first component describing the maximum possible variance. The PCA analysis was implemented using orthogonal rotation with a varimax method for a clear separation of factors. The data were not normally distributed, so a logarithmic transformation of the variables was used. The Kaiser–Meyer–Olkin (KMO) Test and Bartlett's test of sphericity provided standards for sample adequacy and for significant differences among variance that should be passed before conducting the PCA [34]. Details are provided in the supplemental materials (SM-2).

k-Means cluster analysis was used to classify users based on their values of the primary variable. The k-means algorithm is a method of vector quantization for cluster analysis in data mining. Given the simple nature of the algorithm, it is one of the widely used classification techniques. It has been used extensively to understand electricity use patterns

and occupancy patterns (e.g., [17,21]). This method separates a dataset containing n observations into k clusters. A centroid of each cluster is determined with other observations assigned to the cluster with the closest centroid [35].

Our extended cluster analysis focused on users who consume the greatest amounts of electricity to identify additional characteristics that may explain different user types or behaviors. Identification of the optimal number of clusters was based on a comparison of F-statistics and *p*-values and a hierarchical cluster analysis (see Supplemental Materials SM-3).

A correlation matrix served to choose the secondary variable to characterize individual electricity use behaviors. Previous analysis considered multiple variables that were discarded due to concerns with multi-collinearity.

Electricity use associated with mini-fridges in the bedrooms was quantified because they are categorized as energy-intensive appliances and we had the mathematical capacity to identify the electricity use signatures of this particular type of appliance. A subsample of the dataset from 2013–2016 was evaluated in a companion paper to identify the refrigerators based on recurring compressor cycle patterns [36]. Power consumption was analyzed at the minute level for 223 bedrooms. To eliminate noise and other human activities that influence electricity consumption, the analysis considered power consumption for Thanksgiving break when little other electricity use was expected. The results from [36] were matched to specific bedrooms in the analysis presented here to assess the relative importance of the mini-fridges to the total bedroom electricity use.

## 4. Results

### 4.1. General Understanding of Overall Electricity Use

The initial analysis of the electricity data considered 312 bedrooms and all days included in this study. Box plots in Figure 1 and details in Table 1 show a wide distribution of daily electricity use among users, days, and different areas of the apartment. The percentile ranges and maximum values shown in Table 1 include outliers that are not apparent in the box plots (Figure 1).

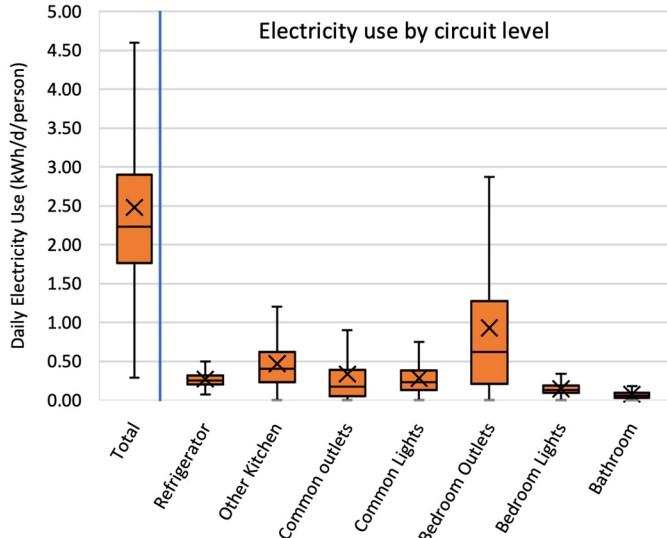

**Figure 1.** Per Capita daily plug load electricity consumption (N = 604 days). Information to the right of the blue line represents use for separate circuits. Bedrooms are the greatest single consumer of electricity. Note: the "×" in the boxes represent the average values and upper whiskers are calculated as Q3 + 1.5 × IQR, where Q3 is the 75th percentile and IQR is the interquartile range.

**Table 1.** Daily per capita bedroom and apartment electricity use distributions, 312 students.

| Percentile | Bedroom-Level Electricity Use | | | Apartment-Level Electricity Use | | |
|---|---|---|---|---|---|---|
| | Daily Use (kWh/d/Person) | Electricity Used by Percentile Range (kWh) | % Cumulative Electricity Used by Percentile | Daily Use (kWh/d/Person) | Electricity Used by Percentile Range (kWh) | % Cumulative Electricity Used by Percentile |
| *50th* | 0.62 | 3029 | 13.3% | 2.23 | 3457 | 34.7% |
| *90th* | 2.24 | 7498 | 64.6% | 3.71 | 3396 | 80.0% |
| *95th* | 3.01 | 3844 | 78.5% | 4.33 | 1379 | 88.0% |
| *99th* | 4.50 | 4291 | 94.1% | 6.47 | 1425 | 96.2% |
| *Maximum* | 13.21 | 1628 | 100% | 20.47 | 649 | 100% |

The median per capita electricity consumption is 2.23 kWh/d/person at the apartment level and 0.62 kWh/d/person in bedrooms. Bedrooms represent the highest electricity use overall (36.6%) and highest variability in use. Unlike other shared living spaces in the apartment, an analysis of bedroom outlets offers the opportunity to attribute electricity consumption to the individual occupant.

The overall magnitude and variability in day-to-day electricity use for all apartments and bedrooms emphasizes that there are days with very high levels of electricity use that are substantial contributions to the overall energy consumption. Of the nearly 30,000 bedroom-days in this analysis, the top 5% of days consume 21.5% of the electricity, and the top 1% consume 5.9%. The nearly 7000 apartment-days similarly show the consequence of outliers, with the top 1% consuming 3.8% of the total electricity. The much higher relative importance of these outliers at the bedroom level (99th percentile = ~7 times the median) versus the apartment level (99th percentile = ~3 times the median) adds evidence to the importance of using granular data to explore individual behavior rather than the more typical approach of measuring aggregated household electricity use. The apartment overall per capita values do not illustrate the same wide behavior among energy users that the individual bedrooms provide.

*4.2. Identifying Primary Measure to Characterize Individual Electricity Use Behaviors*

Evaluation of time series graphs for bedrooms helped to identify individual behavior contributing to the wide variability of electricity use shown in Figure 1 and Table 1. Figure 2 provides samples of the hourly and daily bedroom electricity use for four individual students. There are clear differences in the magnitude of these students' electricity use both among students and within each individual's own habits. Daily use ranged from 0.2 to 5 kWh, with some students using relatively constant electricity from day to day, others had random spikes in their use. Similarly, the hour-to-hour behavior is also quite different among the students, with some having very high electricity use at night, but they manage to turn at least some of their loads to sleep. In all cases, the highest loads were at night, though the magnitude of those loads varied considerably. Median hourly loads at midnight for these four students range from 0.02 to 0.25 kWh/h.

We further explored the dataset to identify additional emerging patterns and specific variables that were most pertinent for understanding individual energy use. That is, descriptive and comparative statistics (Wilcoxon test) were applied to bedroom electricity usage to explore and identify key variables that affect electricity use patterns. This involved analyzing overall electricity usage and consistency of electricity use over 24-h periods to understand peak usage patterns or other kinds of identifiable sequences (e.g., Figure 2). Relative electricity use between bedrooms and common areas were compared. Other factors explored included the year, apartment size (3-, 4-, 6-member apartments), time considerations such as variations over months, classroom scheduling (Tu-Th versus M-W-F class schedules), and potential differences between weekends and weekdays. The analysis showed that these additional variables did not explain differences or patterns in electricity use and could be ignored as relevant parameters across the dataset.

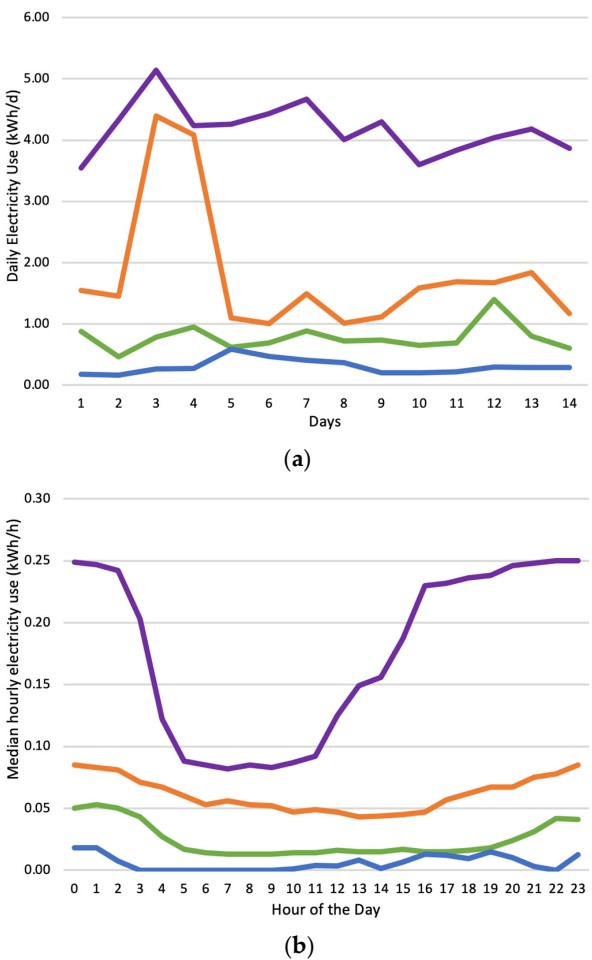

**Figure 2.** Example electricity use for four representative individuals (each a separate color) illustrates a wide range of the magnitude and variability in their electricity use on (**a**) day-to-day (arbitrary 14-day sample) and (**b**) hour-to-hour (as represented by median values for each hour for each individual).

Ultimately, two general categories (Figure 2) emerged as the most likely key variables to demonstrate critical differences in individual electricity use behaviors: the magnitude and variability of electricity use on both hourly and daily time scales. These variables are consistent with the work of [21,22] who identified the magnitude and variability (or heterogeneity) in electricity use behavior as key factors for understanding individual contributions to a household's consumption and targeting those with a higher responsibility for the use for energy efficiency interventions.

Table 2 lists the single-point aggregated measures considered to quantify these characteristics. Due to their non-normal distributions, logarithmic transformation of each of these measures and percentiles were used. Additional normalized measures to represent daily variability (e.g., (75th − 25th)/50th)) were also considered. However, the median electricity use was zero for too many individuals, preventing use of the median in the denominator to normalize measures.

A principal component analysis determined the most influential measures included in Table 2 that describe the magnitude and variability in individual consumption. The PCA analysis showed that median daily electricity use (log transformed) was the top loading component, explaining 49% of the variance in the data (additional details in the supplemental materials, SM-2).

**Table 2.** Single-point aggregated measures to quantify individual's electricity use behaviors.

| | Daily Measures | Hourly Measures |
|---|---|---|
| Magnitude | • Median daily<br>• Max daily<br>• Min daily | • Max of the median of each hour<br>• Min of the median of each hour |
| Variability | Based on percentiles of daily elec. use:<br>• 75th–25th<br>• 90th–10th | Based on percentiles of median elec. use of each hour of the day:<br>• 75th–25th<br>• 90th–10th |

A k-means cluster analysis meaningfully classified users according to the measure that best expressed the magnitude of each individual's electricity use: the log transform of the median daily bedroom electricity use. Two sets of different cluster possibilities emerged (three vs. four clusters). We ultimately chose the set of four clusters as the optimal model (F = 1083, *p* = 0.000) for a four-cluster model vs. three-cluster model (F = 938, *p* = 0.000). The final k-means cluster analysis classifies users into four unique clusters: misers, moderate users, hogs, and super hogs. Additional details are included in the supplemental materials, SM-3.

Figure 3 organizes residents in increasing value of their median daily electricity use. Color-coding identifies the individuals in each cluster. The median of the median per capita daily electricity consumption for super hogs (2.17 kWh/d) is twelve times the median of median daily electricity consumption for low users (0.18 kWh/d). More than 50% of the individuals consume only relatively low or moderate amounts of electricity, but the top 18%—the "super hogs"—account for 48% of total electricity use. This shows that a small number of individuals can have a tremendous impact on electricity consumption.

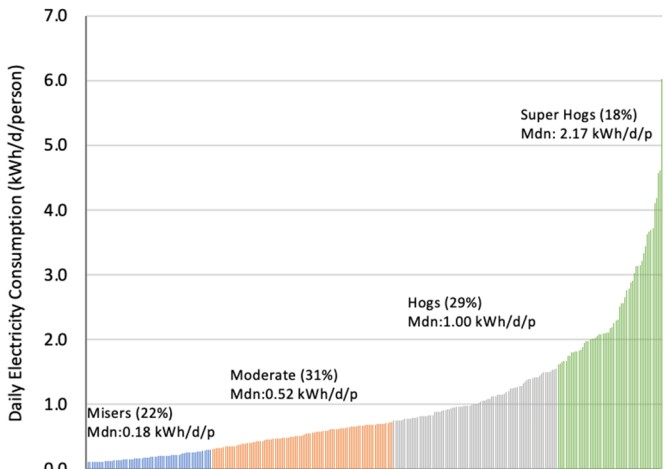

**Figure 3.** The daily median electricity for each resident is shown in ascending order. Users are segmented into four categories based on cluster analysis of their (log) median daily electricity use. The percentages quantify the proportion of the 312 individuals associated with each cluster.

*4.3. Secondary Measures to Characterize Individual Electricity Use Behaviors*

The magnitude of daily electricity use is the primary variable to characterize electricity use behavior and cluster users. However, the wide variability of behaviors within each of these individuals (e.g., Figure 2b) was not captured by this primary variable. Hourly and daily variability was assessed by several measures defined in Table 2. The measure quantifying the 75th–25th percentiles of the daily electricity use explained 14% of the variance in the PCA, and the 90th–10th explained 11%. However, a correlation analysis showed that both of these variables were highly correlated to the primary variable, so they

were not considered further here as a means to further characterize individuals' behavior. In contrast, the measure that describes variability in the hour-to-hour electricity use by each individual defined by the differential between the 90th–10th percentiles of the median electricity use for each of 24 h per day (log transformed) was the least correlated with our main variable (r = 0.559). These hourly patterns give us insight into the variable behavior within these high energy users. As illustrated with four individuals in Figure 2b, some individuals exhibit a large range in hourly electricity consumption, often with minimal consumption in early mornings. In contrast, others show constant electricity use over the day. There is clearly a difference in how residents think about and use electricity.

The secondary measure, which characterizes hour-to-hour electricity use based on the 90th percentile (~highest use hour) and 10th percentile (~lowest use hour) was used to further differentiate electricity use behaviors for individuals in each of the high-energy user clusters. Cluster analysis enabled the hogs and super hogs to be sub-divided into 3 and 4 sub-groups, respectively (Figure 4).

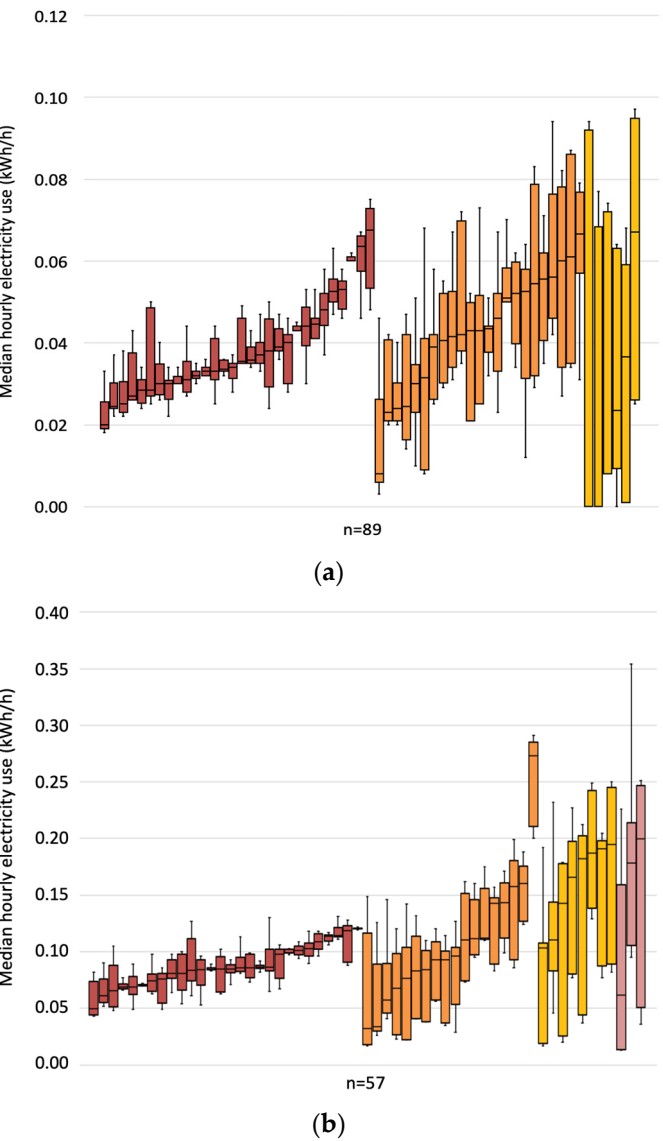

**Figure 4.** Hog (**a**) and super hog (**b**) clusters were further divided by the characteristic that measures hour-to-hour variability. The sub-clusters are identified by color and arranged left to right from the clusters with the least to the most hour-to-hour variability. Within each sub-cluster, box plots that characterize variability in each individual's hourly electricity use are arranged in order of increasing median hourly use.

Even among the super energy hogs—who represent only 18% of residents—we find users that present different patterns in their behavior. Some super hogs have hour-to-hour variability that is extreme, ranging as much as 0.2 kWh/h over the course of a day, whereas the hourly electricity use varies by less than 0.01 kWh/h for other super hogs. Within each of these sub-clusters, there is little correlation between the magnitude of their use and the hour-to-hour electricity use behaviors. For example, the second sub-cluster for the super hogs (Figure 4b, orange) includes both the lowest and highest median hourly use, and the sub-cluster with the highest variability includes one individual with a median hourly use that is fifth lowest among all of the super hog individuals.

There are many implications and suppositions that one can make about these residents. For instance, a user with high electricity use that has small hourly variability may have appliances on all the time such as a computer server or a mini-fridge. In contrast, a user with high electricity use at specific times during the day may be doing time-specific activities such as playing video games, amplified guitar music, high-intensity limited-time computing, or even regular use of a hair dryer. This variability in the magnitude and variation in individuals' electricity use is consistent with what has been reported in office situations [25,26] and household energy use [21].

### 4.4. Appliance Analysis

Variations in electricity use in bedrooms between residents and within individual's hour-to-hour patterns clearly indicate that there is a range of different appliances and electronics used by, and used differently by, the population considered here. Occasional apartment walk-throughs of the apartments and bedrooms in this study show that mini-refrigerators and TVs are the most common appliances in the bedrooms, followed by fans, computers, and gaming systems. Three amplifiers, a recording studio, and one air conditioner were also observed in one effort to document appliances. Fans do not consume much electricity, although the other electronics and appliances can contribute to the substantially high electricity loads of the hogs and super hogs.

In previous work using the Smart Housing testbed, Gao et al. [36] used electricity use patterns to detect the presence of refrigerators in bedrooms using data from 2013–2016. Most other appliances do not have the necessary power signatures for reliable detection, so our analysis here was limited to refrigerators. Gao et al.'s results showed a total of 72 mini-fridges detected out of the 223 possible cases (32%). Results were confirmed by visual inspection of the rooms during mid-semester holiday periods (with student permission). Their analysis demonstrated that mini-fridge electricity use remained similar over the semester.

This mini refrigerator analysis was integrated into the analysis presented here to understand the contributions of this appliance to high electricity consumption in student bedrooms. Of the 72 refrigerators defined by Gao et al. [36], 66 of them were in the 195 bedrooms included in the present study (34%) for the 2013–2016 period. Mini-fridge electricity use ranged between 0.16 to 0.97 kWh/d with an average usage of $0.45 \pm 0.16$ kWh/d. For comparison, most current Energy Star certified mini-fridges (1.6–3.2 cu. ft.) use 0.41 to 0.55 kWh/d, with no consistent trend between energy consumption and size [37]. Overall, mini-fridges contributed to 33% of the electricity use in bedrooms that had these appliances, and 16% of all bedroom electricity use.

Mini-fridge use varied with living situation and the overall electricity use within each bedroom. They were found more often in six-person apartments (56% of the fridges), than in four-person apartments (44%), which is to be expected based on the higher number of people sharing a single kitchen refrigerator in the larger apartments. As shown in Figure 5 and Table 3, a higher percentage of super hogs (64%) have mini-fridges compared with the hogs (46%) and moderate use (27%) clusters. However, among the residents in the moderate use clusters, the refrigerators make up a greater fraction (64%) of the total electricity use in bedrooms with mini-fridges, suggesting that there were fewer other appliances or electronics in these bedrooms. There were no mini-fridge users in the lowest

electricity use cluster. These trends and the diminishing importance of mini-fridges to the total electricity use by the super hogs is evident in Figure 5.

    We initially expected that the presence of the 24-h/d electricity consumption by refrigerators would result in lower hour-to-hour electric load variability for the individuals who had refrigerators. Additional analysis of the sub-cluster of higher use individuals defined as hogs, however, shows that this expectation is only weakly affirmed (Figure 6). The 2013–2016 subset of the values included in Figure 4a now also identifies those with refrigerators (hatched shading). As expected, the individuals with the greatest variation in their hour-to-hour electricity consumption (yellow bars on the right), had no refrigerators. However, many people with no refrigerators had very little hourly variability and some with refrigerators had a great extent of hourly variability.

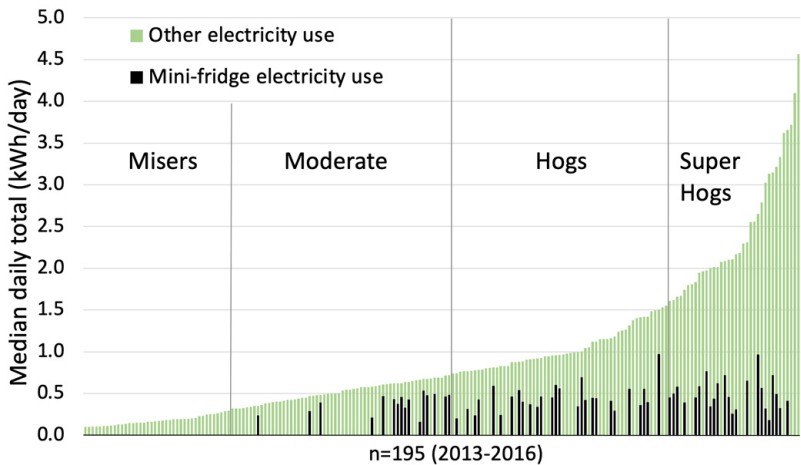

**Figure 5.** Median daily bedroom electricity use among 195 residents (2013–2016). Dark bars indicate the contributions from mini-refrigerators. The super hogs have a higher percent of residents with refrigerators, although refrigerators do not contribute substantially to the total electricity use of these individuals.

**Table 3.** Electricity use by mini refrigerators—analyzed by cluster.

| Cluster: | Miser | Moderate | Hogs | Super Hogs |
|---|---|---|---|---|
| Number of bedrooms | 40 | 60 | 59 | 36 |
| Number with a fridge | 0 | 16 | 27 | 23 |
| Percent of bedrooms with a fridge | 0 | 27% | 46% | 64% |
| Avg. fridge elec use (kWh/d) | 0 | 0.39 | 0.45 | 0.50 |
| % fridge elec use—mini-fridge owners | 0 | 64% | 44% | 21% |
| % fridge elec use—overall bedrooms | 0 | 20% | 20% | 13% |

    Overall, the mini-refrigerator analysis was not sufficient to understand the magnitude or variability of electricity consumption, especially of the hog and super hog clusters. Super hog users typically consume 2–5 kWh/d in their bedrooms (median across all days for each individual), with a few individual days as high as 13 kWh/d. A review of standard appliance power and energy use [38] was considered here to envision what it might take to reach these values. At the median of all super hogs (2.2 kWh/d, Figure 2), daily gaming activity for ~7 h could reach this energy consumption (200 W gaming console + 100 W 50 inchLED monitor). For those super hogs at the higher end of the consumption (median daily electricity use ~4.5 kWh/d, Figure 2), reaching this level of electricity use could be accomplished by adding an inefficient refrigerator (1 kWh/d, Figure 5) and increasing the power consumption of the gaming computer to 400 W, still with the 50inch LED TV. Gaming activity would be required for ~7 h on average every day. Room-sized heaters or air conditioners could also contribute to the high electricity use values we observed.

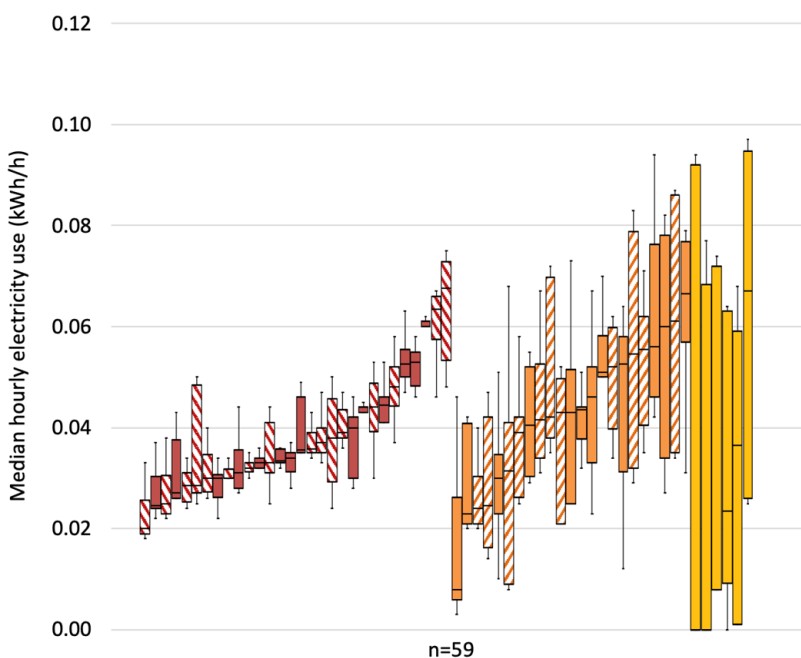

**Figure 6.** Hourly variation of electricity use of hogs separated by sub-cluster. Individuals with hatched shading indicate that they had a mini-refrigerator in their bedroom.

*4.5. Value of Intervention or Policy Initiatives to Reduce Electricity Consumption*

Analysis of highly granular data (hourly and daily measures at the circuit level) allows policymakers, administrators, or building managers to consider policies or interventions to address specific behaviors. This is especially needed in housing such as rental units, or university or military housing, where the residents are not influenced to change their behavior based on a monthly electric bill [11].

Four hypothetical policy outcomes are discussed here in terms of their potential to reduce electricity use. The potential savings are based on the energy use distributions presented in Table 1 and refrigerator use summary statistics in Table 3.

1. No mini-refrigerators (16% reduction of bedroom electricity use);
2. Median daily bedroom electricity use for an individual (across the semester) does not exceed the 90th percentile of use (2.09 kWh/d; 33% savings of bedroom electricity use);
3. Bedroom daily electricity does not exceed the 90th percentile as determined across the semester, all 312 bedrooms (2.24 kWh/d maximum, 35% savings of bedroom electricity);
4. Apartment daily electricity use per capita use does not exceed the 90th percentile as determined for all apartments, all days (3.71 kWh/d/person; 20% savings of apartment electricity use).

Mechanisms to move toward these outcomes could include daily electricity use feedback with motivational messages (e.g., climate change, health benefits, social norming), significant fines for excessive consumption, or an outright ban on refrigerators. Approaches such as a monetary fine are consistent with deposits in rental units or end-of-semester charges to students for damage to the housing unit. Excessive electricity use could be considered in an analogous way to other forms of damage, and some universities are moving towards sub-metered apartments. The collection and interpretation of granular electricity consumption data would justify and facilitate any of these approaches. It is the intent here to explore the potential benefits if such reductions could be achieved, not to develop specific policies that would be effective to reach these outcomes.

Extrapolating the results obtained from residents in the Smart Housing Test Bed to all 689 (per year) apartment residents at Clarkson University helps to show the potential broader impacts of these changes. We estimate that the apartments on campus consume ~75 MWh electricity per year, with 28 MWh of those in bedrooms (analysis details are included in the supplemental materials, SM-4). This represents just 0.3% of the main campus' total electricity consumption. Of the four potential policy or intervention outcomes, they could potentially save 4.4 (no refrigerators) to 15 (reduce apartment daily electricity) MWh electricity per year. Focusing on the bedroom reduction in electricity could save 9–10 MWh/y. These savings are not additive as reductions in any one of these would affect other measures too.

In the context of one university, these savings are small. However, there are also potential co-benefits for the students living in these apartments. Through any energy conservation interventions or policies used to achieve these reductions, these young adults will learn the consequences of their electricity use potentially both through external motivation prompts (fines or bills) or internal motivation through feedback, messaging, and social norming [8]. Any retention of these motivating factors could carry into their post-collegiate lives where additional savings could accrue.

The key findings of this work can be generalized well beyond this particular test bed. The general nature of the results, which show that there is a wide range in the magnitude and variability in how individual residents use electricity, are likely typical of other situations where the residents are not directly responsible for paying their electricity bills and thus do not have a financial motivation to adopt more conservative behaviors. Other groups in comparable situations include young people in many mid-latitude, developed countries, multi-family dwellings, army barracks and other group living environments. In these cases, we can expect a small fraction of individuals who consume a disproportionately large fraction of the electricity resource, but with very different use behaviors and patterns among them.

While generalizability may extend to the broader population, this paper does not address cross-cultural differences in different countries. For instance, European or Asian cultures may have a greater inherent willingness to conserve energy than students in the United States and may consume less electricity than shown here.

## 5. Discussion

In this study, individual electricity consumption is quantified at a highly granular level. Previous studies have focused only on electricity use at the apartment or household level but have not provided insights on individual electricity use among members living in the same house. Our analysis shows that bedrooms are the area in student apartments that present the highest electricity use and high variability among individuals. The understanding of individual electricity use at the bedroom level provides insights into different type of users, their patterns of consumption and impacts of appliances on individual electricity use. These findings provide opportunities to design policies and interventions that specifically address these types of patterns and behaviors such that energy conservation and efficiency goals can be more successfully implemented.

The analysis of 312 users resulted in identification of 4 types of users: misers, moderate, hogs and super hogs. The unequal distribution of electricity use among type of users shows that 18% of users are responsible for 48% of electricity consumption overall. This is consistent with the Pareto Principle analogy that states that for many phenomena about 80% of the consequences are produced by 20% of the causes [39].

The analysis of the impact of mini-fridges on electricity use highlights the need to better identify and assess the role of appliances on electricity use. Asensio and Delmas [24] emphasize that electricity use linked with appliances and electronics have been rising in recent years. Furthermore, their study found that the two major contributions to electricity use at the household level are linked with plug load at 36% and refrigerator use at 19%. Likewise, Ref. [40] draws attention to the need to identify and target specific appliances

based on contribution to electricity use. Thus, it is relevant to understand the dynamics around the use of electronics among individual members of the house to provide insights for setting better strategies towards energy conservation.

From a policy perspective, the use of granular data that provides information of electricity use at the individual level can help to improve policies that use information-based strategies to promote energy conservation. Individual profiles can assess users' characteristics with more precision to better design and improve the impact of interventions for specific subpopulations [20] that rely on information strategies to promote energy conservation. Even though our study analyzes the impact of only one major appliance, future studies can work on understanding other energy-intensive appliances such as high-powered electronics to understand the appropriate focus of interventions and whether interventions should be more efficient if they target a one-shot specific action such as upgrading refrigerators or if it is necessary to target the development of daily habits such unplugging electronic equipment when not in use.

From a supply and demand management perspective, understanding specific profiles can help to target users that have significant contributions to peak load [22]. Asensio and Delmas [24] emphasize the importance of load shifting behavior from peak hours to off-peak hours to increase reliability in the power grid system and reduce the risk of blackouts, brownouts, and other failures. Several of the papers discussed here mostly use mathematical algorithms to identify particular energy use activities, hourly load profiles, and individual behavior. They generally do not have substantial databases of measured electricity use data to verify their approaches (e.g., [19]) or truly understand the nature of incentives that would help to shift or decrease peak loads. In contrast, our measurement and use of granular data at the individual level can provide insights into electricity use, including variability among residents' magnitude of use and peak hours.

This study showed the value of granular electricity data for understanding and quantifying key variables that differentiate individual electricity use behaviors, but it is not without limitations. Individual energy use in this study assumed that circuit-level data for single bedrooms in student apartments are a good proxy to study individual electricity use. Figure 7 illustrates the value of using bedroom data to assess individual electricity use behavior relative to the aggregated apartment-level data. These three apartments are representative of many living situations where individuals within apartments have very different personal electricity use behaviors. Information for all apartments is included in the supplemental materials (SM-5). For the second and third apartments (apt. quartiles 2 and 3), the presence of individuals in the super hog cluster (#4) would not have been identified if only the apartment's overall quartile was used as a measure of the behavior of all residents. In contrast, using only the bedroom data to assess behavior of individuals in the first apartment (quartile 4) did not capture their likely high electricity use measured in the common living areas. The two individuals with the * designations had low bedroom use but in an apartment with high use in common areas. While the use of only bedroom data is an acknowledged limitation of the work presented in this paper due to its lack of inclusion of electricity use in other living areas, a relatively small number of individuals (26 students; 8.3% of the 312 total) were identified who were likely falsely identified as electricity use misers.

| Cluster # | Apt. Quartile | Electricity Use (kWh/d/person) | | | |
|---|---|---|---|---|---|
| | | Bedroom | | Common Outlets | Apartment Total |
| 4 | 4 | | 2.10 | 2.31 | 5.75 |
| 1 ✳ | 4 | | 0.20 | 2.31 | 5.75 |
| 2 ✳ | 4 | | 0.56 | 2.31 | 5.75 |
| 2 | 3 | | 0.45 | 0.12 | 2.45 |
| 4 | 3 | | 2.90 | 0.12 | 2.45 |
| 3 | 3 | | 1.14 | 0.12 | 2.45 |
| 1 | 3 | | 0.15 | 0.12 | 2.45 |
| 2 | 2 | | 0.68 | 0.05 | 2.08 |
| 3 | 2 | | 1.34 | 0.05 | 2.08 |
| 4 | 2 | | 1.74 | 0.05 | 2.08 |
| 1 | 2 | | 0.22 | 0.05 | 2.08 |
| 4 | 2 | | 2.51 | 0.05 | 2.08 |
| 1 | 2 | | 0.14 | 0.05 | 2.08 |

**Figure 7.** Bedroom and apartment per capita electricity use for 15 students living in three separate apartments. The cluster assignment for each individual (1 = misers through 4 = super hogs) is compared to the quartile assigned to the apartment total electricity use (2 = 25th–50th percentile though 4 = 75th–100th percentile). The median electricity use is shown for the individual (blue bars are a fraction of the maximum, 6.0 kWh/d); the apartment common living areas (green bars relative to the maximum, 2.31 kWh/d/person) and the overall apartment total electricity use (orange bars relative to maximum, 5.75 kWh/d/person). The * identifies two individuals for whom the bedroom electricity use might not be a good proxy for their individual electricity use behavior due to the very high electricity use in the common area.

## 6. Conclusions

The present research contributes to the scientific understanding of residential individual electricity consumption. Our results demonstrate that high users who consume a disproportionate fraction of the total electric energy can be divided into distinct subgroups based on variation in the magnitude of their hourly load profiles, with some high users showing small hourly variability and other high users with significant variability in electricity use). Residents' lack of information about their electricity use creates a gap between the degree to which people acknowledge the need for resource efficiency practices and their actual conservation actions [10,11]. Thus, this basic knowledge about the dimensionality of users, which has not previously been published, can be leveraged to better target or align intervention and policy to specific types of users and behaviors. Previous work has focused on measurement in simpler situations (e.g., offices) or use of mathematical algorithms to disaggregate data into component parts. Our measured data for a large number of residents provides an opportunity to test and validate these modeling approaches.

The general trends that we see in student usage are likely generalizable to broader populations who also are not directly motivated by the receipt and payment of monthly electricity bills. While this is an untested proposition, the trends of high variation in usage magnitude and hourly variation (for instance, late evening peak, midday peak, or constant high-usage) are very likely a trend that we might see more broadly with other young relatively affluent adults who are in situations where they do not directly see or pay utility bills.

Future work based on the smart housing testbed correlates these individual electricity use characteristics with their respective responses to a survey of energy use attitude and motivations to examine additional underlying reasons for high-use behavior among some residents.

**Supplementary Materials:** The following supporting information can be downloaded at: https://www.mdpi.com/article/10.3390/su15054171/s1. SM-1. Basic Descriptive Statistics, Table S1. Individual Bedroom and Days per Resident Data; Table S2. Daily bedroom electricity use for each cluster. SM-2. PCA analysis, Table S3. Correlation Matrix for variables included in PCA; Table S4. KMO and Bartlett's test statistics; Table S5. PCA Communalities; Table S6. PCA Total variance explained.

SM-3. SPSS Results Cluster analysis, Table S7. Clusters Defined; Figure S1 Hierarchical clustering—validation of clusters. SM-4. Details—analysis of annual savings if policy outcomes achieved, Table S8. Details—analysis of annual savings if policy outcomes achieved. SM-5. Analysis – Value of individual measures and bedrooms as proxy for individual use, Figure S2e Median electricity use of all residents organized by apartment. SM-6. Demographic characteristics dataset, Table S9. Distribution of student majors in dataset.

**Author Contributions:** C.B.: methodology, formal analysis, data curation, writing—original draft; S.B.: conceptualization, writing—review and editing, supervision; L.L.: conceptualization, formal analysis, supervision; S.E.P.: conceptualization, writing—review and editing, visualization, supervision. All authors have read and agreed to the published version of the manuscript.

**Funding:** Funding for this phase of this research was provided by internal Clarkson University sources. Funding for the establishment of the Smart Housing testbed was provided by Clarkson University and the New York State Research and Development Authority, NYSERDA Contract 32013, and also via support from IBM Inc.

**Institutional Review Board Statement:** The study was conducted in accordance with the Declaration of Helsinki, and approved by the Institutional Review Board of Clarkson University (Protocol # 14-07, 09/27/13) for studies involving humans..

**Informed Consent Statement:** Informed consent was obtained from all subjects involved in the study.

**Data Availability Statement:** The anonymized circuit-level hour and daily electricity use for all days of the seven year (fall semesters) study period are accessible through Mendeley Data digital commons, https://doi.org/10.17632/53pjfj84d6.1.

**Conflicts of Interest:** The authors declare no conflict of interest.

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
