# Peer review of "Energy Hogs and Misers: Magnitude and Variability of Individuals’ Household Electricity Consumption"

_sustainability, doi:10.3390/su15054171_

Round 1
Reviewer 1 Report
circuit-level electricity measurements, this keyword is too long
Line 57, SeeChange 2020, wrong citation?
Line 63, what is granular data?
Line 95, outlet data is incorrect?
Cite some recently published electricity paper:
https://www.mdpi.com/2071-1050/14/11/6650
Objective one, Understand, describe, and quantify individual electricity consumption at a granular level. The author should tell readers what is granular level before this objective, one sentence will be sufficient.
Objective 2, Identify variables to characterize individual electricity consumption behaviors. It seems this is known to everyone? What is the value of this one?
Objective 3, Outline the benefits of these findings based on achieving interventions and policies outcomes. This one seems to cover too many things, we need a narrower objective.
The research presented here focuses on student behavior in their bedrooms? Why? Which university? I guess culture differences make different research outcomes.
Line 144, what kind of engineering majors (66%)?
3. Methods should be Research Methods
Point forms should be turned to paragraph form.
Lines 180, 189, 242, 244, 261, 281 etc delete the unnecessary space after full stop.
Line 188, disqualified should be excluded?
Copy edit title of Table 1, line 317, A user with high electricity use at specific times in line 356
Line 261, delete in this case
Figure 3 is not cluster analysis
Labels of Figure 5 and 6 should be shortened to at most 1.5 lines
Line 431, what do you mean by highly granular analysis?
Section 4, even if we know their electricity habits, so what is the academic contribution? If we can know different groups like engineering students use more electricity than arts students, even they are taught in class, that will be of some academic interests. In this present context, we cannot see any practical, academic or policy values now in the results section 4.
Similar problem in Lines 475-476, Our analysis shows that bedrooms are the area in student apartments that present the highest electricity use and high variability among individuals.
Reviewer 2 Report
The topic of the manuscript is relevant, and its content may be of interest to potential readers. However, there are lot of gaps and shortcomings that authors need to remove before a manuscript can be published:
1. Regarding the structure of the document:
1.1. The last introduction paragraph should describe the manuscript structure. It is advisable to start this paragraph as follows:
"Further material is divided into several parts. Thus, in Section 2……. Section 3 presents……".
1.2. A certain drawback of this document is the lack of a separate section with a literature review. This review was mainly given by the authors in the introduction, but it could be more extensive.
1.3. Section 2 should probably be better called "Objectives of research and materials". Then the title of subsection 2.1 could be deleted.
1.4. Section 3 has subsection 3.2. However, I don't see the title of subsection 3.1.
1.5. It is bad that the text of the article ends with a picture (Figure 7). I recommend adding section 6, which should present conclusions, limitations, and prospects for further research.
2. Regarding the content of the manuscript:
2.1. It seems to me that some of the tables, which are included in the supplementary materials, should be placed in the main part of the manuscript. This is especially true for the first two tables.
2.2. Overall, in my opinion, Table SM-1.2 is one of the most important in this study. And I believe that the authors did not fully reveal the differences in the amount of electricity consumption by different groups of consumers. These differences are very significant. What are they caused by?
2.3. It would be desirable to provide a fairly complete list of factors that may cause these differences. Why did the authors limit themselves mainly to mini-fridges? Was it not possible to conduct an analysis of all household electrical appliances?
2.4. How to distinguish the necessary volumes of electricity consumption from excessive ones? What factors led to the lack of an energy-saving behavior model among some consumers?
2.5. The authors did not sufficiently clearly explain the proposed mechanisms for stimulating the reduction of energy consumption. In particular, financial incentives are not entirely clear. Please note that if this article is published, it will be read by readers from many countries. Not all of them know how payment for electricity consumption occurs in the student housing studied by the authors. Perhaps it would be appropriate to explain this in more detail.
3. Regarding the grammar, style and design of the manuscript text:
3.1. Some sentences could be worded better. However, the main point concerns the style. The style is not always sufficiently scientific. In particular, this applies to the annotation. It is good that the authors present the quantitative results of their research in the abstract. However, it is also necessary to clearly state the objectives of the study, the methods and the object of the study.
3.2. Authors should carefully revise the text of the manuscript to eliminate typos and technical errors (for example, in line 506).
3.3. It is necessary that the design of the manuscript text meets all requirements. In particular, all abbreviations in the text and symbols in the formula should be deciphered at the first mention. Check that references are correct (maybe you need to specify the source numbers in square brackets?).I think it is appropriate to acquaint the authors with these comments, suggestions and questions. I hope that such acquaintance help to improve the quality of the manuscript, which is expected to be published in such a high-ranking journal as "Sustainability".
Reviewer 3 Report
Congratulations on the interesting, exhaustive and well-structured and visualized article as well as for the realized organization of the implementation of household electricity consumption measurements.
I have a recommendation to include in the title of the article the word "student", since the research is aimed at expanding the details for this type of individual electricity consumers, and not for Household Electricity Consumption in general.
I have some small notes on the layout of the text:
- it is good to avoid the transfer between the pages of the subfigure inscription in Figure 1 and the lines of Table 3;
- there is one unnecessary transition of the text to a new line in the sentence of lines 505-506;
- I notice unnecessarily large spaces between "." and the beginning of the next sentence in a number of places in the text - lines: 282, 321, 433, 459, 462, 523, 526, 528, 529...
Round 2
Reviewer 1 Report
The author should check again all the comments in the review
Line 11 abstract, what kind of and statistical analysis
Line 14, cluster analysis may be a wrong description as it most likely refer to those text mining stuff
Line 15 18% of students consume 48% of total electricity use
Our analysis contributes to a deeper understand- 23 ing of the different ways in which individuals use energy, which allows for targeting interventions 24 to groups with similar patterns of consumption, this contributions may be restricted to the US and probably even worse, to that universities only. Can the results be generalized?
Lines 25-26, An analysis of the benefits of reducing excessive 25 electricity use shows potential savings ranging from 16-33% of bedroom electricity., why do we need to know percentage like this?
Line 49information deficit and line 55 wastefulness are incorrect terms?
Line 59, what is granular data?
Lines 95-97,The study finds 20% are high energy consum- 95 ers, 60% medium energy consumers and 20% low energy consumers. However, this 96 work did not examine whether (or how) these groups differed beyond overall usage. Why do we need to know about this?
To understand and assess individual residential electricity consumption, the Smart 109 Housing testbed at Clarkson University (Potsdam NY, USA) was developed. Why did the author select this uni? Can the results be generalized?
It is known that weather plays a very important role in electricity usage pattern due to hot summer needs air conditioners and winter needs heaters. Yet, this study failed to consider the weather conditions. It is unclear why this unit selected without telling what makes it special
Extend conclusion part by stating the academic, practical and policy implications of the paper
Testbed Description in section 2 heading is strange. We usually do not use this keyword for heading.
Research objectives are rarely used in a section as well.
Line 117, profile individuals, profile is a wrong word here.
The objectives have to be rewritten, say the first one, describe cannot be an objective:
Understand, describe, and quantify individual electricity consumption at a granular level in residential settings.
Is granular level an appropriate term?
Identify and improve our understanding of variables linked with individual-level 124
energy consumption in residential settings. What variables do you refer to? could you please state that directly?
Lines 128-129, uses plug-load data to focus on student behavior in their
bedrooms. What is plug load data?
Phase 1. Preliminary analysis to identify trends, potential variables, and limita- 158
tions in the dataset:We normally state the variables based on literature.
State what is k-means cluster and application per https://www.sciencedirect.com/science/article/abs/pii/S1470160X18301298
Line 245, how do u find these students?
State what is Wilcoxon test with formula.
Title of figure 5 should be kept in 2 lines only. Similar problems for others, like Figure 7
Line 418, what is hogs’ cluster?
Add a few journal articles from 2021-2023.
Conclusion is too short. It should state its academic, practical and policy contributions and limitation and future research direction
Reviewer 2 Report
In my opinion, the text of the manuscript has improved. The authors took into account a number of my comments. The authors provided reasoned explanations for some of my remarks. However, the manuscript still contains some flaws. The main disadvantages include the following:
1. The authors did not fully reveal the reasons for the significant differences in the energy consumption of different consumers. The authors point out that data on electrical appliances, except for refrigerators, is difficult to obtain. But wouldn't it be possible, for example, to conduct a survey of residents? In general, I understand that, probably, the authors will try to take this remark into account in their further work.2. The authors did not explain the mechanism of payment of expenses for the used energy, which is used in this particular case. Residents do not pay each separately? Do they have no financial incentives to reduce energy consumption? These questions arise for me in connection with the author's proposals set out, in particular, in lines 463-465: "Mechanisms to move toward these outcomes could include daily electricity use feedback with motivational messages (e.g., climate change, health benefits, social norming), significant fines for excessive consumption, or an outright ban on refrigerators”. Do the authors consider these proposals realistic?
3. The scientific novelty of the research results needs to be checked. In lines 488-491, the following is noted: "In this study, individual electricity consumption is quantified at a highly granular level. Previous studies have focused only on electricity use at the apartment or household level but have not provided insights on individual electricity use among members living in the same house”. Is this statement fair? For example, in line 108, the authors note the following: "This analysis is some of the first to do so." That is, such studies were still conducted?
Other comments:
1. I cannot understand why the authors did not take into account my remark 1.1 from the previous review. The description of the structure of articles in the last paragraph of the introduction is quite common in this journal (see, for example, https://www.mdpi.com/2071-1050/15/3/2751).
2. Please check the order of arrangement of figures and tables according to references in the text. In particular, it seems to me that figure 5 should be placed after table 3.
3. It is not desirable to include explanations of figures in their names. It is better to transfer these explanations to the text of the article.
4. In my opinion, grammar and style still need improvement.
Round 3
Reviewer 1 Report
The authors should revise according to reviewers’ comments.
Polish English
Abstract, policies that might reduce the excessive electricity use, what policies?
Line 45, [5–9], please state what did these citations said.
Line 50, information deficit is an incorrect collocation.
Lines 54-57, split incentive problem, means that , because residents…potential for conservation. It is unclear about this key term’s definition, please replace by a simpler definition.
Lines 61-62, what do you mean by “electricity data, which are disaggregated, high frequency measures of electricity at the circuit level”?
At the end of section 1, state what are the remaining sections. Move This manuscript includes four additional sections…strategies to the end of section 1. It only needs sections 1, 2, 3 but not those decimal points.
Line 12, why do the author highlight “disaggregated data at the circuit level with high frequency measurements”?
Line 139-140, why “The research presented here uses data from appliances, electronics and lights 139 plugged into outlets to focus on student behavior in their bedrooms”?
2.2, what is Testbed Characteristics?
Why the sample of Clarkson’s Smart Housing was selected?
Line 165, what are reflects outlets and hard wired lighting?
190, Wilcoxon test needs citation and explanation.
Line 191, Principal Component Analysis needs citation
Cite where have 190, 191 and clustering tests have been applied, say, clustering was applied in Modularity clustering of economic development and ESG ...
Maths formular for PCA etc should be included.
Where have K-means clustering been used? Please state A spatio-temporal analysis of low carbon development in China’s 30 provinces: A perspective on the maximum flux principle
Line 248, Electricity use associated with mini-fridges…, we all know that many fridges in the modern day has different electricity label to see whether it can save energy. As such, we may need to know the level of energy saving in each of the dormintory’s room? Yet, it also implies that using different types of electrical appliances seriously affect electricity. So, what is the point of including this study of student hall when some students use different brand?
Shorten the label of Figures 1-4.
Line 292, why bold?
Line 397, I still cannot see the “There are many implications”, it only refers to a student hall, different halls may have different habits, some like play at night time obviously use more electricity. Some have habits of not turning off the light, some hall has smart lighting etc tha turn off the unused lighting.
The paper is not very convincing at present form why we need this paper.
Line 404, bold is not needed.
Lines 521-528, unclear
